



# Three-year database of atmospheric measurements combined with associated operating parameters from a wind farm of 2MW turbines and rotor geometry

Caroline Braud[1,2], Pascal Keravec[1], Ingrid Neunaber[1,7], Sandrine Aubrun[1], Jean-Luc Attié[3], Pierre Durand[3], Philippe Ricaud[5], Jean-François Georgis[3], Emmanuel Leclerc[3], Lise Mourre[4], and Claire Taymans[6]

[1]LHEEA lab. (ECN - CNRS), 1 rue de la Noe, 44300 Nantes
[2]CSTB, 11 rue Henri Picherit, 44300 Nantes
[3]LAERO, Université de Toulouse, UPS, CNRS, 14 Avenue Edouard Belin, 31400 Toulouse, France
[4]VALOREM, 213 cours Victor Hugo, 33323 Begles Cedex
[5]CNRM, Université de Toulouse, Météo-France, CNRS, 42 Avenue Gaspard Coriolis, CEDEX, 31057 Toulouse, France
[6]VALEMO, 213 cours Victor Hugo, 33323 Begles Cedex
[7]NTNU, Department of Energy & Process Engineering, Norwegian University of Science & Technology, NO-7491, Trondheim, Norway

**Correspondence:** Caroline Braud (caroline.braud@ec-nantes.fr)

**Abstract.** A comprehensive meteorological dataset from an operational wind farm, consisting of six 2 MW turbines, has been made available. A meteorological mast, equipped with sonic anemometers at four different heights, was installed at the center of the farm and has collected data over three years. The dataset is further supplemented with radiometer measurements for atmospheric stability analysis. Simultaneously, supervisory control and data acquisition (SCADA) data were acquired to pro-

5  vide operational information about the wind turbines, including inter alia power production and wind direction. Additionally, the turbine blades were scanned to support aerodynamic simulations. This unique and comprehensive database has been made accessible to the research community through the AERIS platform.

## 1 Introduction

In this work, we present a database of atmospheric measurements within a wind farm. The database contains environmental

10  data collected over a 3-year period by a meteorological mast (met mast) and a radiometer, both located near an onshore wind farm consisting of six Senvion MM92 wind turbines. Additionally, supervisory control and data acquisition (SCADA) data from four of the six turbines are included for the same period. These datasets together with rotor and blade geometry provide essential information on the operating states of the turbines, enabling the assessment of wind turbine wake dynamics and the associated wake-induced turbulence, either through physical models or numerical simulations.

15  Depending on wind direction, the met-mast is exposed either to undisturbed atmospheric flow or to flow affected by the wakes of the turbines. The met-mast is equipped with four sonic anemometers, which allow for detailed measurement of wind speed components, as well as accurate assessment of turbulence and thermal covariances, even in wake-affected conditions.





The originality of the present database is multifaceted:

- **Operational Wind Turbine (SCADA) Data**: Access to operational data from commercial wind farms is rare, despite the existence of some publicly available databases (Passos et al., 2017; Plumley, 2022; Fraunhofer, 2022). Typically, academic researchers must negotiate agreements with wind farm operators to access such data, with the dissemination of results often subject to industrial approval. By providing this database as Open Data, we aim to attract the attention of researchers who require full-scale data on turbulence properties and wind turbine operations to validate physical and numerical models at both rotor and wind turbine scales.

- **Measurement of Wind Properties**: The wind energy industry typically measures wind properties using cup anemometers or lidar profilers (Duc and Simley, 2022). These sensors, however, are limited in their ability to capture the turbulence tensor and assess the thermal stability of the atmosphere. These limitations are addressed in the present database through the use of sonic anemometers, which enable more comprehensive measurements.

- **Expansion of the Database**: This initial database serves as the foundation for further datasets generated in the same wind farm environment. These additional datasets will be collected through two French research projects, ePARADISE and ANR MOMENTA, for which data are available on the AERIS website https://awit.aeris-data.fr/. Future measurements will include data from instrumented unmanned aerial vehicles (UAVs) and a scanning LiDAR, complementing the current database.

- **Applications and Future Benchmarking**: Some outcomes from this database are already being used by project partners to perform and validate physical and numerical simulations at both rotor and wind turbine scales. When published, the results will provide a valuable basis for broader benchmarking, fostering collaboration with other institutions globally.

## 1.1 Description of the site and of the farm arrangement

The site under investigation is located near Saint-Hilaire-de-Chaléons in western France, approximately $10\,\mathrm{km}$ east of the Atlantic coast and $32\,\mathrm{km}$ west of Nantes (see Figure 1(a)). It consists of six Senvion MM92 wind turbines, each with a rotor diameter of $D = 92\,\mathrm{m}$ and a hub height of $h_{hub} = 80\,\mathrm{m}$, as further detailed on in Section 4. The turbines are arranged in two rows, each containing three turbines, with a row-to-row spacing of $1.2\,\mathrm{km}$ (see Figure 1(b)). The distance between turbines T1 and T2, and between turbines T2 and T3, is approximately $350\,\mathrm{m}$ (equivalent to $4D$), while the spacing between turbines T4 and T5, and between turbines T5 and T6, is approximately $280\,\mathrm{m}$ (equivalent to $3D$).

A met mast, with a height of $79\,\mathrm{m}$, is positioned between the two turbine rows. It is described in Section 2. The distances between the turbines and the met mast are shown in Figure 1(b). The coordinates and terrain elevation $h_{NN}$ at each turbine location and the met mast are provided in Table 1. The terrain in the area is relatively flat, with elevation variations of approximately $\pm 1\,\mathrm{m}$ between the met mast and turbines T3–T6 (see Figure 2). However, turbines T1 and T2 are located $4\,\mathrm{m}$ higher than the met mast.



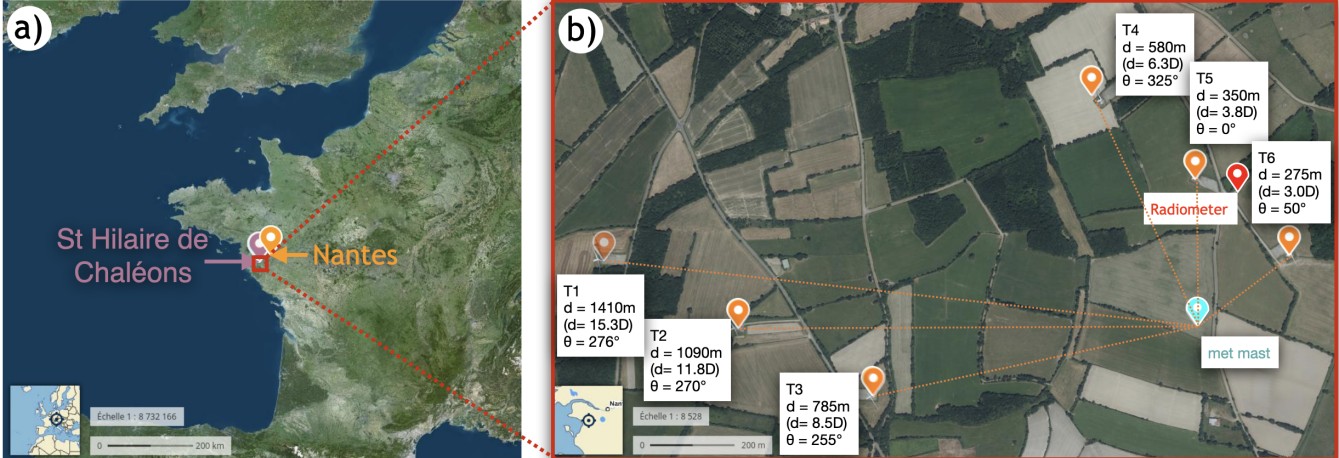

**Figure 1.** Overview of the measurement site (b) at Saint-Hilaire-de-Chaléons in the west of France (a). The site is 10 km from the coast and 32 km from Nantes. The site consists of six wind turbines with different distances $d$ from the met mast (b). Aerial photo source: ©IGN – « BD ORTHO® ».

**Table 1.** Coordinates and elevation above sea level $h_{NN}$ of the turbines and the met mast.

| Turbine | Longitude / ˚W | Latitude / ˚N | $h_{NN}$ / m |
|---------|---------------|---------------|--------------|
| T1 | -1.9243 | 47.0911 | 25.81 |
| T2 | -1.9200 | 47.0896 | 25.12 |
| T3 | -1.9158 | 47.0882 | 22.65 |
| T4 | -1.9087 | 47.0944 | 20.43 |
| T5 | -1.9057 | 47.0928 | 20.85 |
| T6 | -1.9028 | 47.0912 | 20.77 |
| met mast | -1.9057 | 47.0897 | 21.40 |
| radiometer | -1.9045 | 47.0925 | 20.58 |

A radiometer that is further described in Section 3 is installed near turbines T5 and T6 (at a distance of 100 m and 197 m, respectively), as shown in Figure 1(b). It is used to capture variability in the vertical structure of the atmospheric surface layer.

Figure 1(b) presents an aerial view of the terrain at the measurement site. The surrounding area is predominantly flat, consisting of grass fields interspersed with single rows of bushes and trees, which reach heights of approximately $10\,\mathrm{m} - 15\,\mathrm{m}$. To the northwest of the met mast, behind turbine T6, there are groves where the tree height is similar to that of the field hedges.

## 2 The meteorological mast

The meteorological mast was installed in the wake of turbine T6 for a north-east wind direction (see Figure 1). It is a three-legged guyed lattice mast, constructed from 26 triangular sections, each with a height of 3 meters and a width of $0.45\,\mathrm{m}$. The




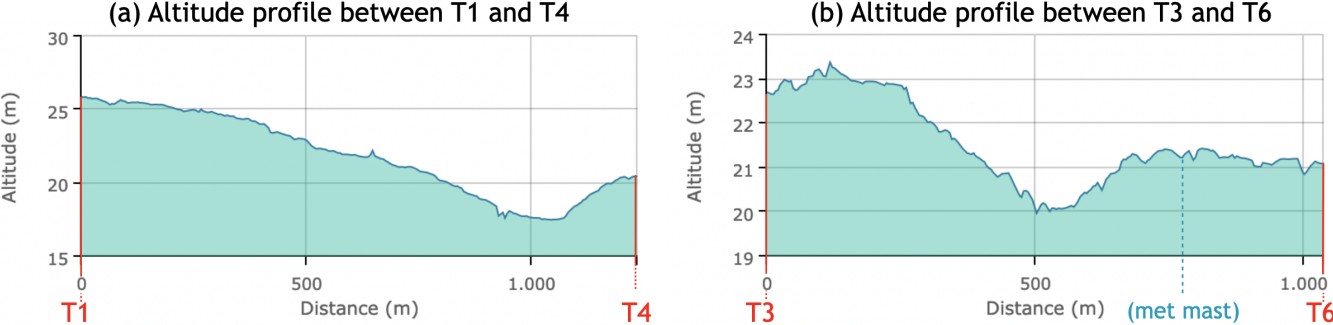

**Figure 2.** Exemplary terrain elevation profiles between (a) turbines T1 and T4, and (b) T3 and T6. The approximate position of the met mast is indicated; note that the met mast is about 75 m away from the line along which the profile was plotted. Digital elevation model source: ©IGN – « RGE ALTI® ».

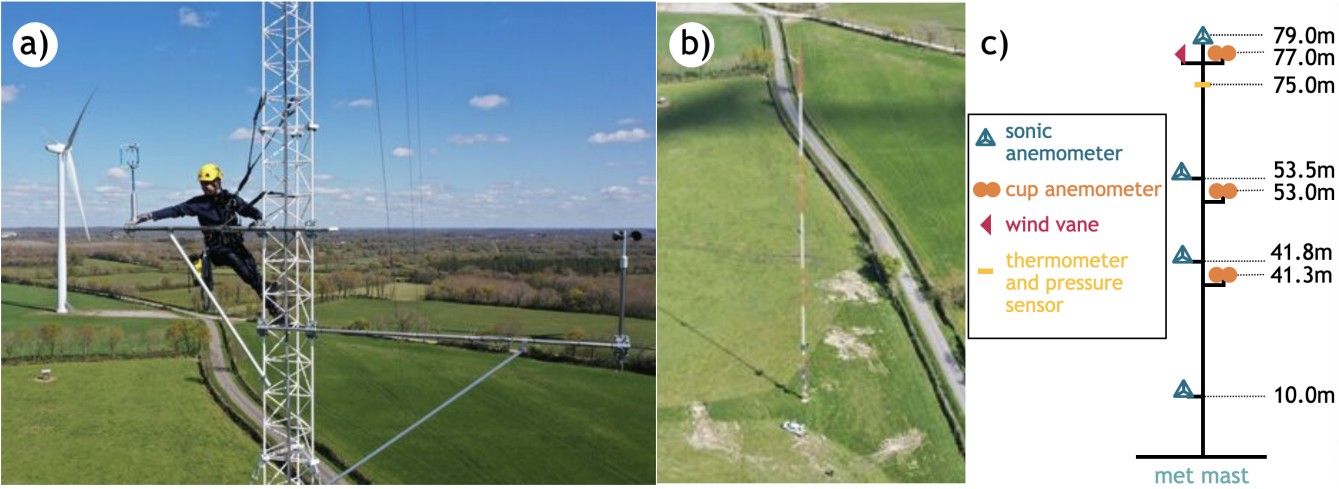

**Figure 3.** Photographs of the met mast: (a) replacement of a sonic anemometer by a rope technician, (b) the met mast, (c) Sketch of the met mast.

guy-wires are attached to the legs of the mast and oriented perpendicularly to the opposite side of each triangular section, at angles of 0°, 122°, and 302°. The guy-wires are fixed at seven different heights (9 m, 18 m, 27 m, 39 m, 51 m, 63 m, and 75 m). The four lower levels are anchored at a distance of 22 m from the base of the mast, while the three upper levels are anchored at a distance of 45 m from the base.

The instrumentation was divided into two main stations: a weather station and a turbulence station. The weather station measures wind speed at three levels (41.3 m, 53 m, and 77 m) using three Thies First Class Advanced anemometers (Thies GmbH & Co. KG, Germany). At the highest level (77 m), a Thies First Class Advanced wind vane is also used to measure wind direction. Atmospheric pressure is measured at 75 m using an AB 60 barometric pressure sensor (Ammonit Measurement



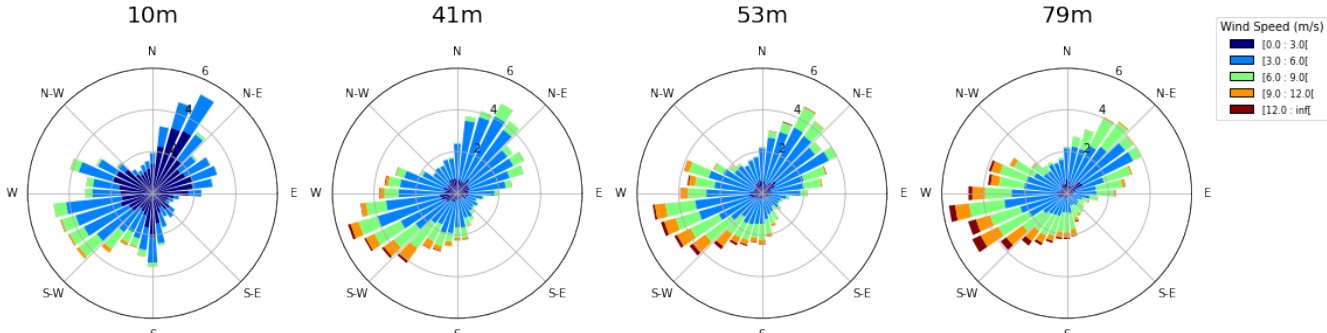

**Figure 4.** Wind roses obtained from sonic anemometers at heights of 10 m, 41 m, 53 m, and 79 m for three years from December 2021 to January 2024.

GmbH, Germany), and air temperature is measured at the same level using a TPC1.S/6-ME thermometer (MELA Sensortechnik GmbH, Germany).

The turbulence station records turbulent velocity components and air temperature fluctuations at four levels ($10\,\text{m}$, $41.8\,\text{m}$, $53.5\,\text{m}$, and $79\,\text{m}$) using four Windmaster sonic anemometers (Gill Instruments Limited, UK). During the field experiment, the top and $41.8\,\text{m}$ anemometers were replaced with two Gill WindMaster Pro sonic anemometers.

Turbulence measurements were collected from December 2020 to January 2024 at a frequency of $10\,\text{Hz}$ to calculate turbulence properties (e.g., turbulence intensities, variances, and covariances) and to perform sensor orientation and calibration corrections, using EddyPro®software[1]. These statistical quantities were computed over an averaging period of 1 hour. Figures 4 and 5 show the wind roses and turbulence intensity, respectively, for the measurement period.

The prevailing wind directions are from the west-southwest and northeast. The wind rose distribution is relatively insensitive
to altitude, with no systematic veer in the vertical wind profile. However, the wind rose at $10\,\text{m}$ shows more scatter compared to the other heights, due to the heterogeneity of the terrain in proximity to the ground. Wind speeds at hub height ($z = 79m$) typically range from $3\,\text{ms}^{-1}$ to $12\,\text{ms}^{-1}$, corresponding to the below-rated operational range of the wind turbines.

The turbulence intensity as a function of wind direction exhibits a similar pattern across all altitudes (Fig. 5). It clearly shows higher turbulence intensity at a wind direction of $50°$, where the wake of turbine T6 is evident. An increase in turbulence
intensity is also observed between $300°$ and $25°$, corresponding to the merged wakes of turbines T4 and T5. The wake effects from turbines T1, T2, and T3 are slightly visible between $240°$ and $285°$. These observations suggest that the atmospheric boundary layer (ABL) turbulence is locally disturbed by wake-induced turbulence from the turbines.

A *wake index* was defined to identify whether measurements from the sonic anemometer are influenced by wind turbine wakes. Green dots represent wind directions where the met mast is not affected by turbine wakes and where statistical con-
vergence is acceptable (i.e., $[90° - 105°]$ ; $[150° - 240°]$ and $[285° - 300°]$). In contrast, black dots indicate wind directions

---

[1]https://www.licor.com/env/support/EddyPro/topics/introduction.html



**Figure 5.** Turbulence intensity obtained from sonic anemometers as function of wind direction at heights of 10 m, 41 m, 53 m, and 79 m for three years from December 2020 to January 2024. Red bars indicate wind turbine locations. Small dots represent hourly data, big dot are the median values by sector.

where the met mast is impacted by wind turbine wakes (i.e., $[0° - 90°]$ ; $[240° - 285°]$ and $[300° - 360°]$) , or where statistical convergence is not acceptable (i.e., $[105° - 150°]$).

Additionally, it is evident that turbulence intensity decreases with altitude for wind directions without wake interactions (freestream conditions). This trend is less apparent for wind directions experiencing wake interactions. The footprint of wake-induced turbulence on turbulence intensity appears to be less sensitive to altitude.



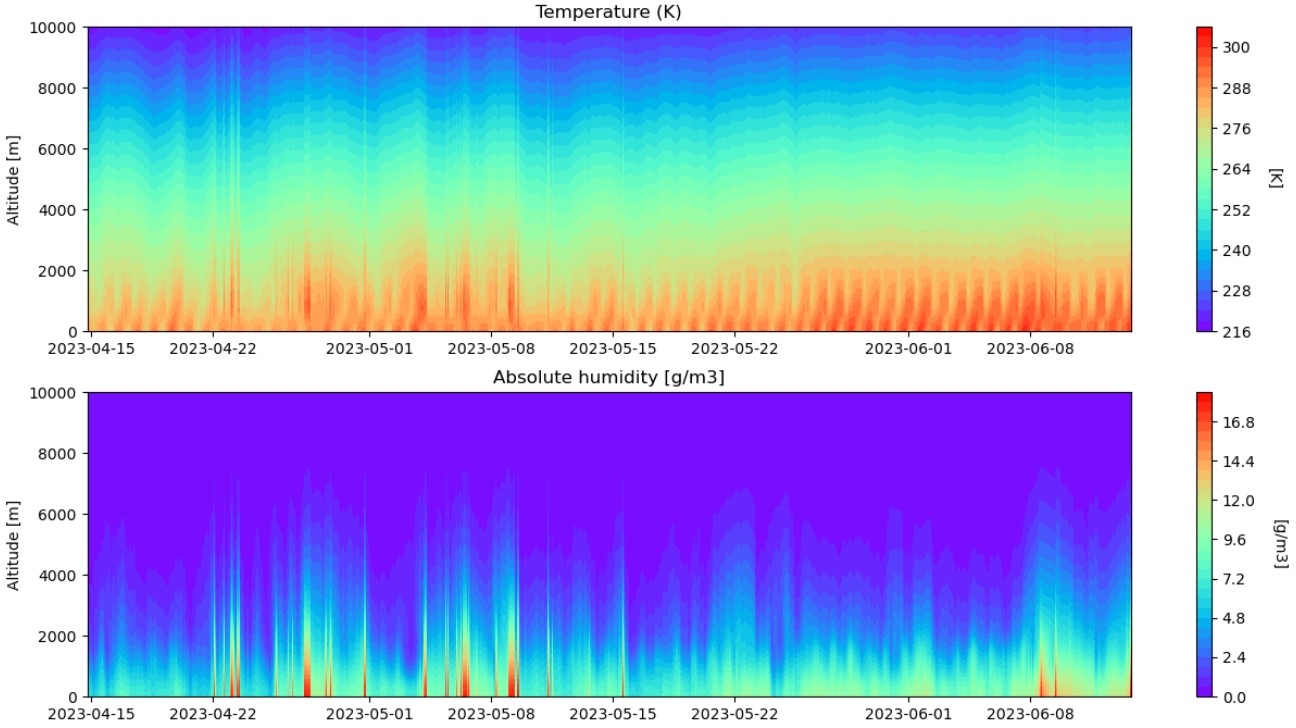

**Figure 6.** Temperature profiles from the K-band (top) and absolute humidity profiles in g/m³ (bottom) measured between 2023-04-15 and 2023-06-15 during the MOMENTA experiment.

## 3 The radiometer

We also operated an RPG HATPRO G2 ground-based passive microwave radiometer (MWR), which has been previously deployed in several field campaigns, including the SOFOG3D (SOuth West FOGs 3D) experiment Martinet et al. (2022). The radiometer measures the sky brightness temperature at various wavelengths, with a radiometric noise between 0.3 and 0.4 K,

using an integration time of 1 second for this experiment. Two frequency bands are utilized by the receivers: the K-band, which targets the water vapor line, with measurements at 7 frequencies (22.24, 23.04, 23.84, 25.44, 26.24, 27.84, and 31.4 GHz), and the V-band, which is sensitive to the oxygen line, with measurements at 7 frequencies (51.26, 52.28, 53.86, 54.94, 56.66, 57.3, and 58.0 GHz).

Using the inversion algorithm (neural network) described by Rose et al. (2005), the radiometer provides retrievals of water

vapor and temperature profiles from the surface up to 10 km altitude, along with measurements of liquid water path and integrated water vapor. The vertical resolution of the profiles ranges from 30 to 50 m in the atmospheric boundary layer, 100 m in the lower free troposphere, and 500 m in the upper free troposphere. Retrievals are performed at 93 levels, with a

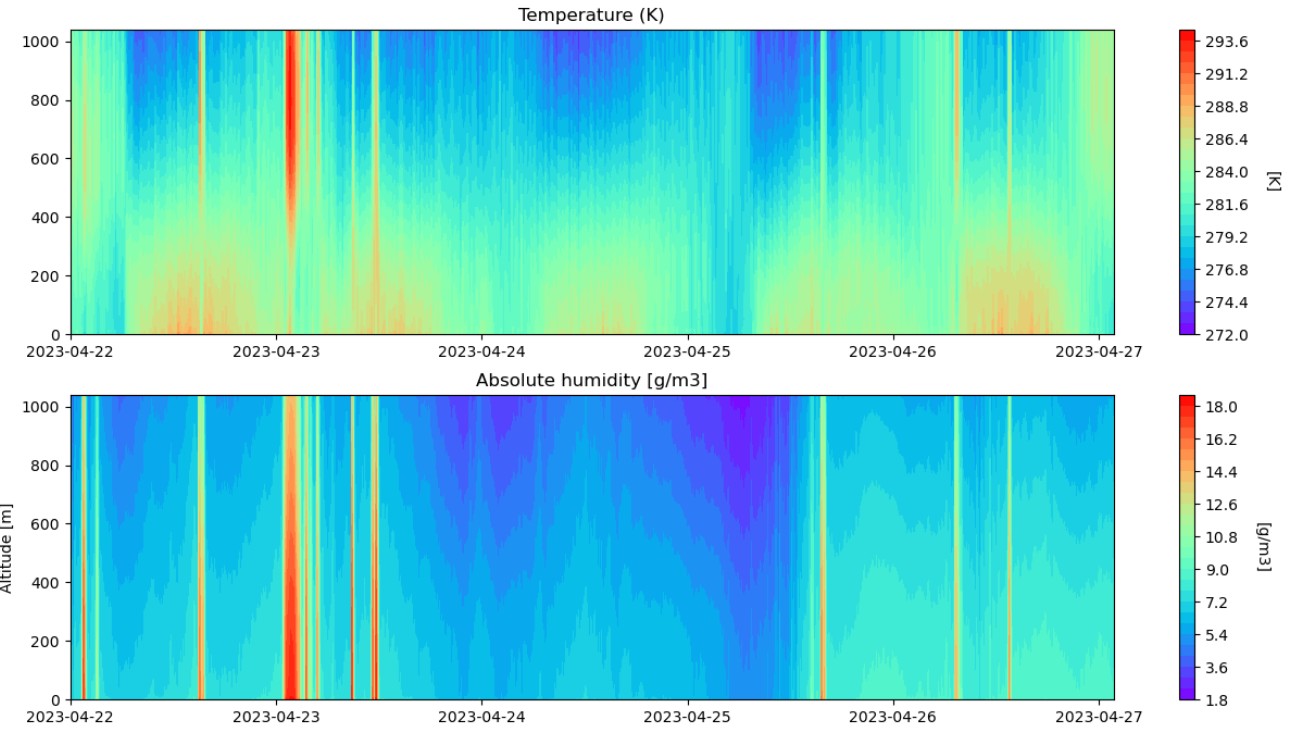

**Figure 7.** Zoomed-in Temperature profiles in the K-band (top) and absolute humidity profiles in g/m$^3$ (bottom) measured in the altitude range of 0-1000 m and between 2023-04-22 and 2023-04-27 during the MOMENTA experiment.

time resolution of 60 s, as outlined by Ricaud et al. (2013). Calibration is performed every 5 profiles, lasting approximately 4 minutes.

This ground-based radiometer was deployed from June 2022 to April 2024 as part of the MOMENTA project in the region located at 47.09254°N, 1.90450°W (Figure 1.) It was positioned near turbines T5 and T6 (at a distance of 100 m and 197 m, respectively) to capture variability in the vertical structure of the surface layer, depending on whether the measurements were taken within or outside the wake of these turbines. Figure 6 presents a sample of temperature and absolute humidity profiles collected between April 15, 2023, and June 15, 2023. A zoomed-in version of these profiles between the surface and 1000 m

altitude is shown in Figure 7, focusing on the period from April 22, 2023, to April 27, 2023. These profiles clearly demonstrate the daily variation of temperature in the lower troposphere, with a dry period observed around April 24th and 25th, 2023, as reflected in the corresponding absolute humidity data. Notably, temperature inversion events (i.e., an increase in temperature with altitude) are consistently associated with the wettest periods.



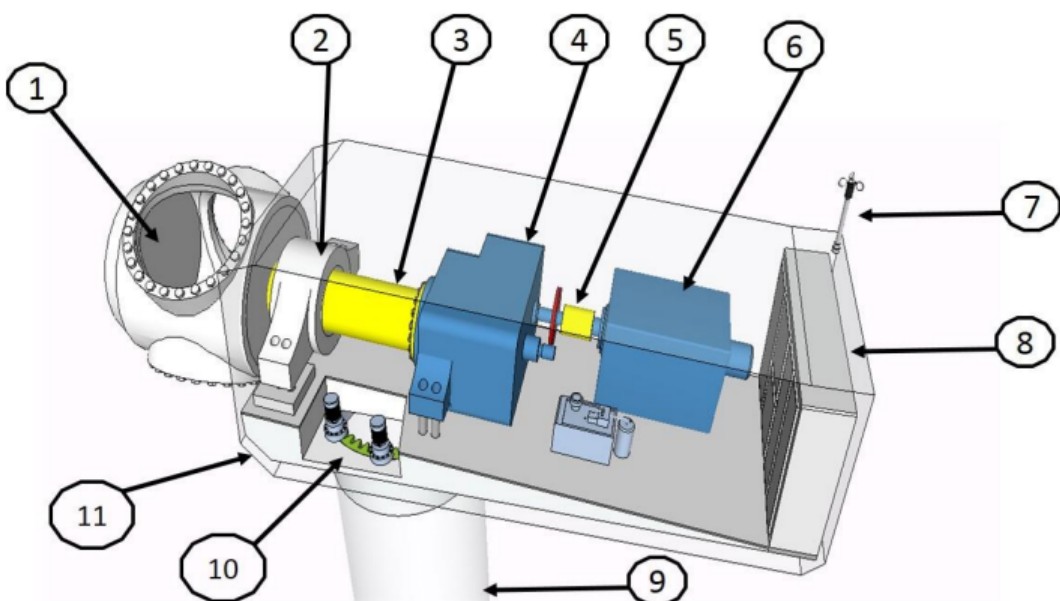

**Figure 8.** Structure of a modern wind turbines (courtesy of Valemo), showing the major components: (1) hub or rotor; (2) main bearing; (3) low speed shaft; (4) gearbox; (5) high speed shaft; (6) generator; (7) measuring anemometer; (8) transformer; (9) tower; (10) nacelle; (11) yaw or azimuth bearing.

**Table 2.** Characteristics of the wind turbines.

| Turbine part | Dimensions |
|---|---:|
| Nacelle height | 80 m |
| Blade length | 45.2 m |
| Tip blade height | 126 m |
| Rotor diameter | 92 m |
| Nominal power | 2.05 MW |
| Rotational speed range | 7.8 to 15.0 rpm |
| Average weight of each blade | 7 979 kg |
| Average weight of the rotor without blades | 17.0 t |
| Average weight of the nacelle without rotor | 69.5 t |

## 4 Description of the Wind Turbine

The wind turbine model used in the wind farm is the MM92, manufactured by Senvion. Figure 8 illustrates the structure of a modern wind turbine similar to the MM92.

The characteristics of the wind turbines are detailed in Table 2.



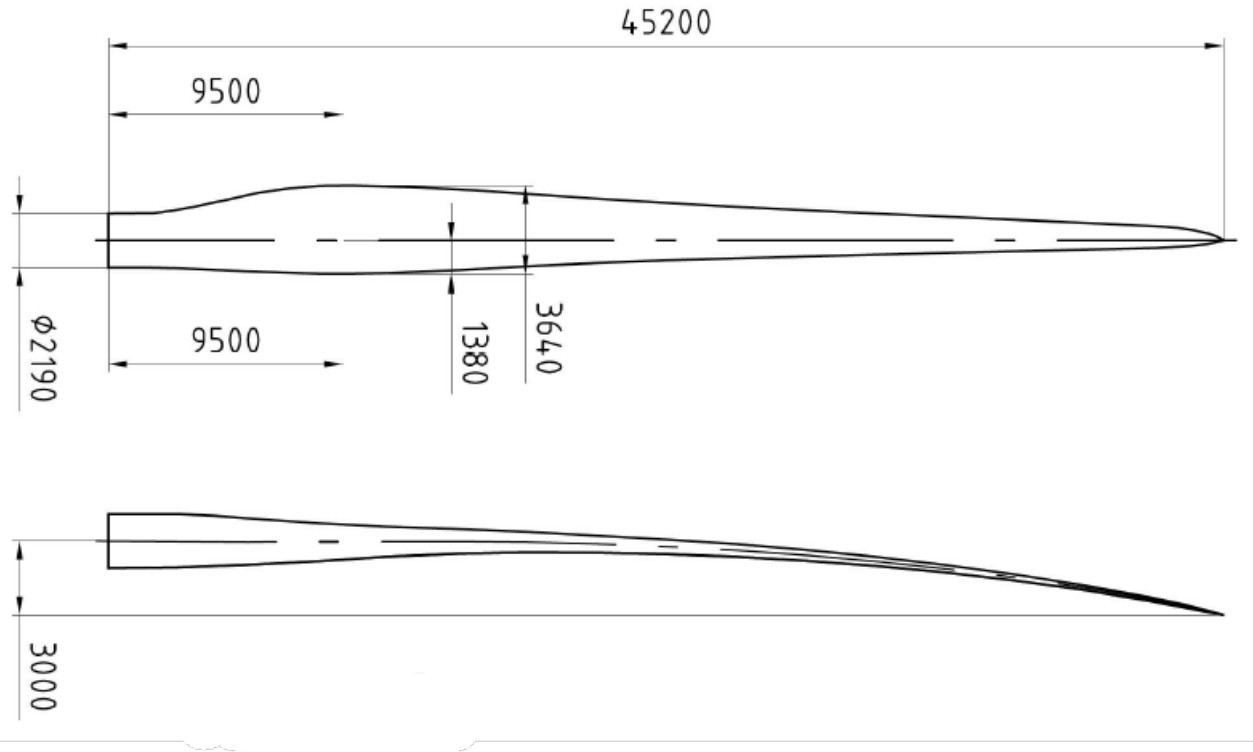

**Figure 9.** General dimensions of one rotor blade (in millimeters).

The blade geometry is also documented through scans described in Section 5. Figure 9 provides the general dimensions of a single blade.

All wind turbines are equipped with a Supervisory Control and Data Acquisition (SCADA) system that records 10-minute averaged data for various parameters. The data collected are categorized into four main groups:

1. **Environmental parameters:**

    – **Wind speed**: Measured using two anemometers mounted on the nacelle, behind the rotor. One is an ultrasonic anemometer with an accuracy of $\pm 0.1\,\mathrm{ms}^{-1}$, used as a reference, while the other is a cup anemometer with an
    
accuracy of $\pm 0.5\,\mathrm{ms}^{-1}$, serving as a backup.

    – **Wind direction**: Determined by combining the nacelle position with the wind vane deviation (measured by the ultrasonic anemometer).

    – **External temperature**: Measured using a PT100 sensor located on the nacelle.

Both wind speed and wind direction are corrected for deviations caused by rotor rotation (which also varies according
to wind speed and azimuth error), although the transfer functions used for these corrections are not provided by the manufacturer and remain unknown.





2. **Electrical characteristics**:

   – **Active power**: Calculated based on voltage and current measurements.

3. **Control variables**:

   – **Pitch angle**: Measured with an encoder located on the blade control motor, with an accuracy of 0.05° for the blade angle.

   – **Low-speed shaft rotation**: Calculated from pulses generated by an inductive sensor activated by a cogwheel.

   – **High-speed shaft rotation**: Measured with an encoder mounted on the generator.

   – **Generated torque**: Derived from the active power measurement and the high-speed shaft rotation.

   – **Targetted Torque**: Targetted torque from the control algorithms.

4. **Status codes and alarms**:

   – These provide detailed information about the operational state of each wind turbine.

## 5  Extraction of the blade geometry

The blade geometry provided by the manufacturer is incomplete. To supplement data, additional scans were conducted. Numerous approaches and tools are available for blade scanning; in this study, two methods were employed: scanning from the ground and during operational maintenance. Both methods were subcontracted to an expert surveyor firm and utilized the same laser 3D scanning technique, which offers an accuracy of $\pm 3$ mm.

Each approach has specific advantages and disadvantages, which will be discussed in detail in Section 5.1. The post-processing method used to generate point clouds for blade geometry extraction is described in Section 5.2. Lastly, the accuracy evaluation of the airfoil shape is presented in Section 5.3.

### 5.1  Scan methods

#### 5.1.1  Scan from the ground

The first scanning method was conducted from the ground, allowing the blades to remain mounted (see Figure 11a). However, several drawbacks were identified:

1. Weather conditions: Despite targeting no-wind conditions, residual wind caused blade movement, increasing the dispersion in the measured point cloud. This variability resulted in the inability to extract a unique blade shape from the data (see Figure 11b).

2. Obstructions: Parts of the blades were obscured by the mast during scanning, creating gaps in the measured point cloud (see Figure 11c). These gaps prevented the complete extraction of the 3D blade properties.




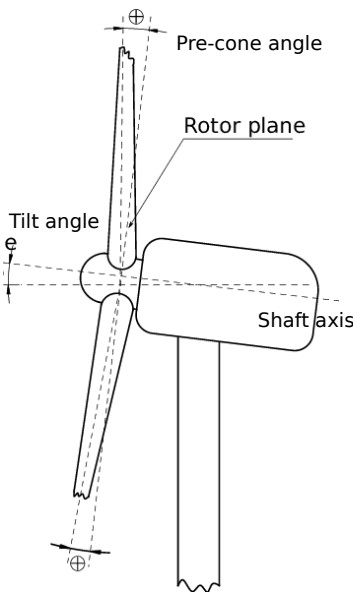

**Figure 10.** Illustration of the pre-cone angle, i.e., the angle between the root blade axis and the rotor plane, and the tilt angle, i.e., the angle between the horizontal axis and the shaft axis. Sketch modified from Menon Muraleedharan Nair (2017).

3. Hub alignment challenges: Determining how the blades were mounted on the hub, including pre-cone and tilt angles, posed additional difficulties (see Figure 10). Evaluating the tilt angle required scanning the entire nacelle with a reference framework from the ground, while extracting the pre-cone angle necessitated knowledge of the rotor plane. Unfortunately, these parameters could not be derived from the data collected using this scanning method. The origin of the hub framework, essential for determining the pre-cone angle, was also inaccessible.

To address these limitations, a method for extracting the pre-cone angle is detailed in the next section, using data from the second scanning method.

Ultimately, one 2D blade section shape at 82% of the rotor diameter was successfully extracted. To evaluate the aerodynamic loads of this blade section, it was extruded, manufactured, and tested in the CSTB wind tunnel (see Neunaber et al. (2022)). Additionally, a 1:10 scale model of this 2D blade section was manufactured and tested in the LHEEA wind tunnel, as well as simulated using URANS equations (see Mishra et al. (2024)).

### 5.1.2 Scan during operation maintenance

Another scan of the wind turbine blades was conducted during a rotor bearing maintenance. Compared to the previous method, this approach presented additional challenges:

1. Blade stabilization measures: Devices such as caps at the blade tip and lanyards were used to prevent blade movement (see Figures 12a) and b). While these stabilizers were necessary, they introduced limitations. For instance, the chosen



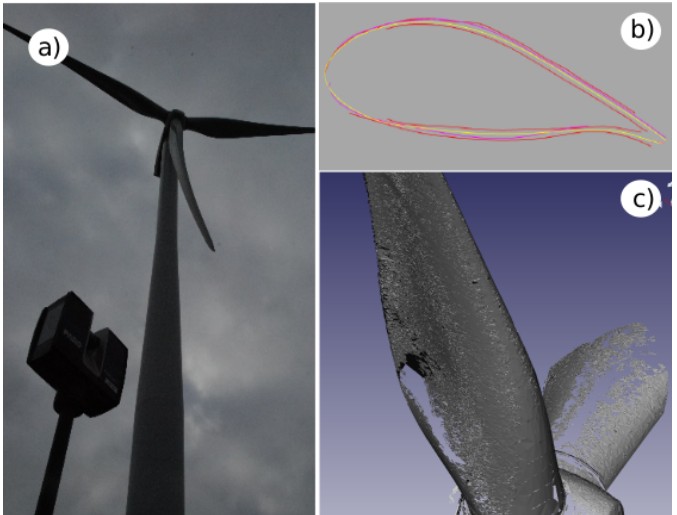

**Figure 11.** First blade scan operation: (a) 3D scanner on the ground close to the wind turbine, (b) scatter of possible blade sections that can be extracted from the cloud of points, (c) missing data ("holes") in the measured cloud of points from this first scan operation.

blade had a lanyard at its tip, which obstructed measurements at this location and hindered accurate determination of the pre-bend angle distribution. Additionally, the lanyard was removed during the scan process to accommodate maintenance, leading to increased dispersion in the measured point cloud.

2. Weather conditions: Similar to the first scan method, low-wind conditions were targeted. However, slight oscillations at the blade tip caused by residual wind during the scan introduced further dispersion in the measured data.

3. Hub insertion and blade length: The blade extends into the hub at the collar location, a region that could not be captured during the scan (see Figure 12c). As a result, the total measured blade length was slightly shorter than the actual length provided by the manufacturer (see Figure 9). The portion of the blade inserted into the hub was estimated to be approximately 0.89 m, as detailed in Section 5.2.

Despite these challenges, this scan method included a complete scan of the hub (see Figure 12d). This allowed the definition of a hub origin, enabling the extraction of the pre-cone angle, as discussed in the next section.

## 5.2 Post-processing of blade scan data

Because the second scanning strategy provided more comprehensive data, it was used to extract the blade geometry, following the procedure outlined below.



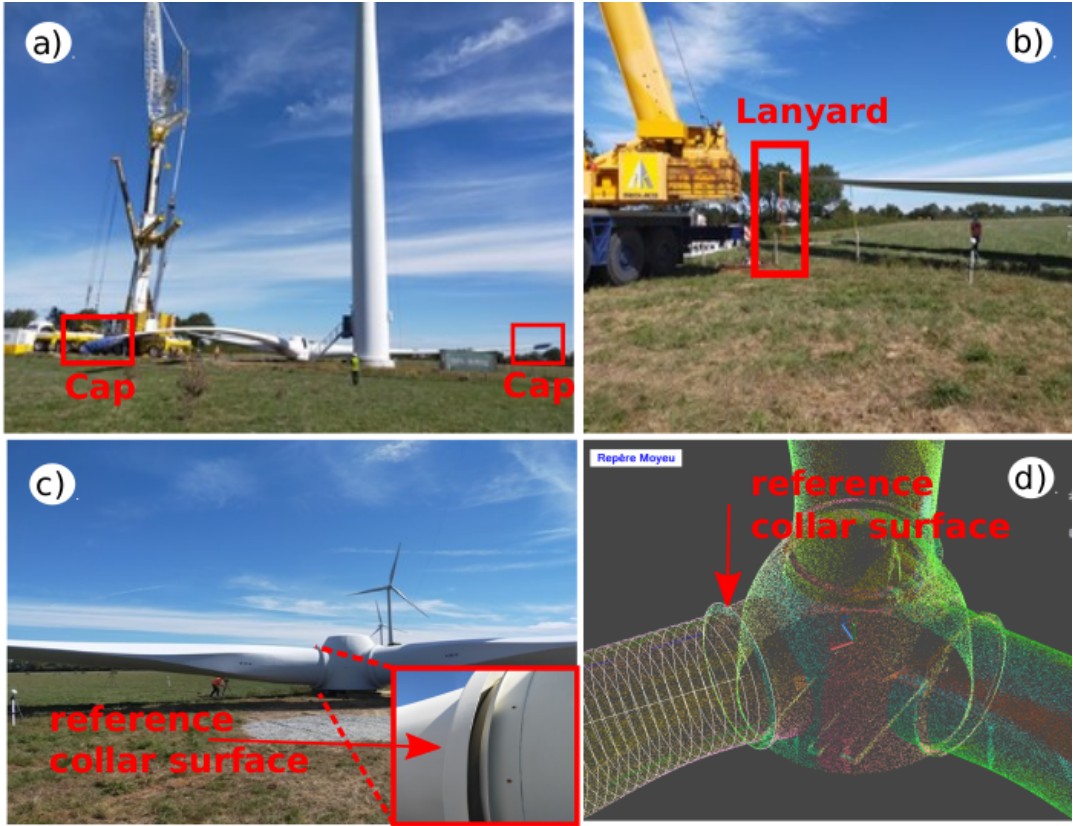

**Figure 12.** Second scan operation: a) photograph of the rotor on the ground with caps at the blade tips. b) Chosen blade for the extraction of the airfoil section; the blade had no cap but was fixed at the ground with a lanyard. c) Zoom of the blade-hub junction marked by a collar. The collar surface on the blade side was used as reference for the origin of the blade framework. d) Measured cloud of points at the hub location with white circles representing the extracted profiles. The first extracted profile was at the reference surface collar on the blade side, with a diameter equal to the collar diameter.

### 5.2.1 Hub and blade frameworks

Blades are elastic and are typically mounted at a predefined angle relative to the rotor plane – the pre-cone angle – to prevent contact with the mast during operation (see Figure 10). To determine this pre-cone angle, the hub framework must first be defined (see Figures 13a) and 13b).

The blade framework originates at the blade-hub junction, with its origin, $O_{blade}$, located on the blade root axis. The framework axes are detailed in Figure 13c).





The distance between the hub and blade origin is 2 m, corresponding to a translation of the hub framework by:

- $dx_h = 0.140$ m,
- $dy_h = 0.043$ m,
- $dz_h = 1.998$ m.

The pre-cone angle was calculated to be 3.8°, which is slightly higher than the manufacturer-specified value of 3.5°.

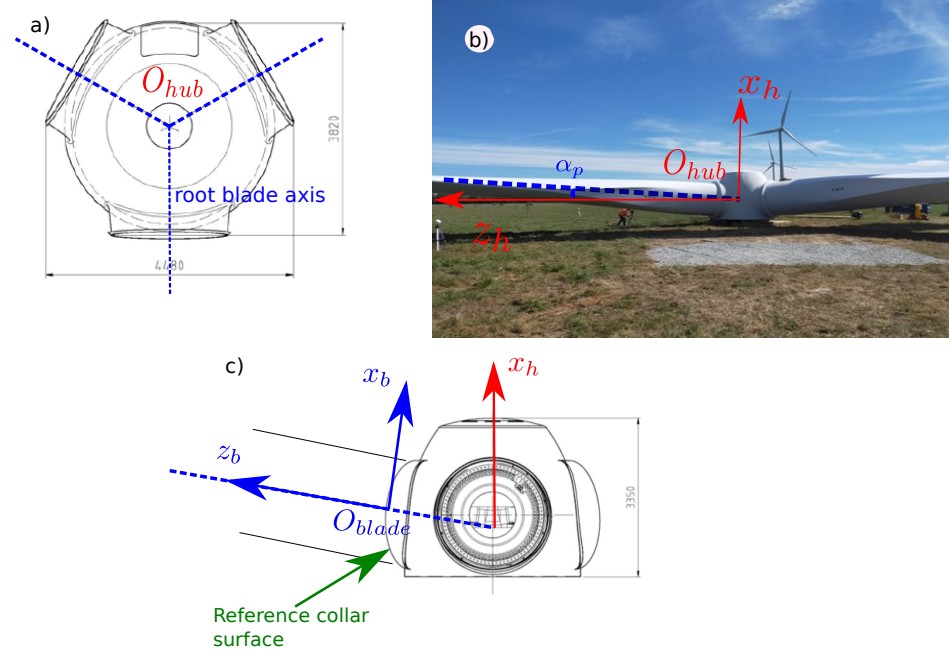

**Figure 13.** Definition of the coordinate framework of the blade (measures are in millimeters): a) Front view of hub. Dotted blue lines indicate the blades axes at the root location. The intersection of the three blade axes defines the hub framework origin, $O_{hub}$. b) The $z_h$ axis is parallel to the ground, $x_h$ is normal to $z_h$ and towards the sky and $y_h$ is such that the hub framework is an orthogonal, right-handed one. $\alpha_p$ is the pre-cone angle, defined as the angle between the root blade axis and the $z_h$ axis. c) Side view of the hub. The blade framework origin is the center of circular blade root section, $O_{blade}$ (i.e. on the blade root axis) and starts at the reference collar surface, $y_b$ is parallel to the chord direction of the largest blade chord (see figure 14), $x_b$ is such that the blade framework is an orthogonal, right-handed one.





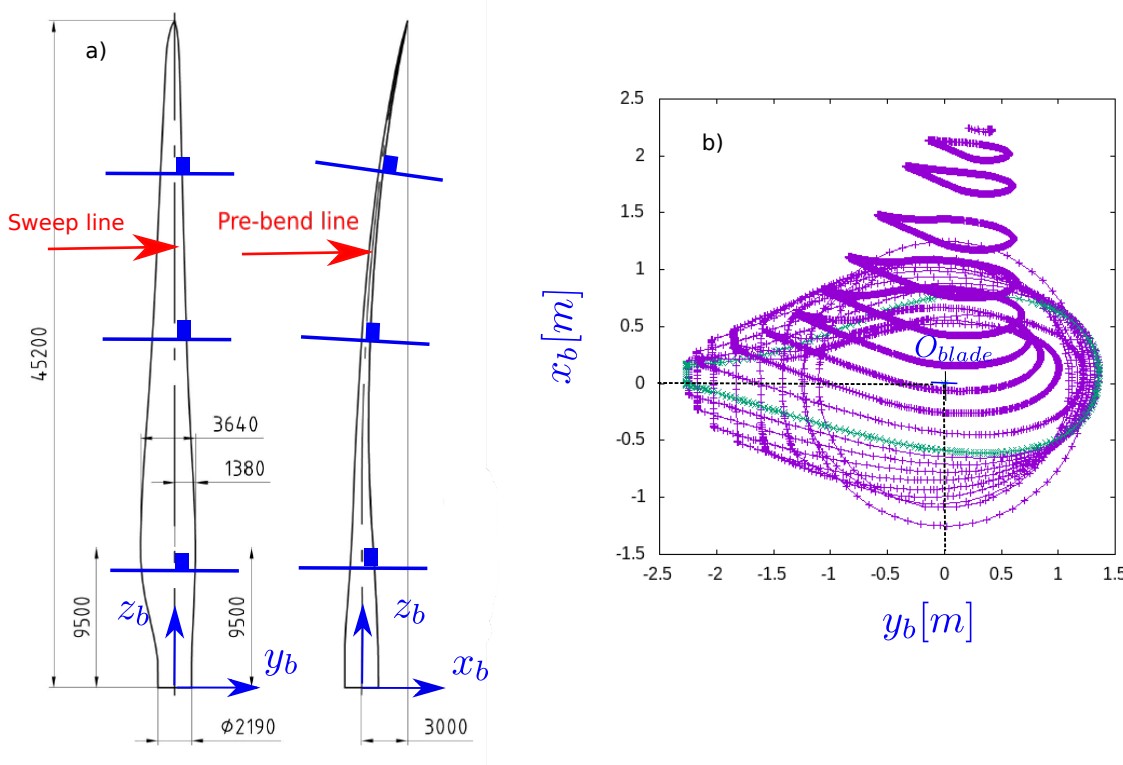

**Figure 14.** Extraction of blade sections (measures are in millimeters): a) extraction of blade section along the sweep and pre-bend lines; b) 45 sections along the blade were extracted. The green section has the maximum chord, and its orientation is used to define the $y_b$ axis.

### 5.2.2 Extraction of blade sections

Once the blade framework was established, blade sections were extracted along the sweep and pre-bend lines (see Figure 14). This process was performed manually: when $z_b$ deviated from the blade root axis, $y_b$ was adjusted along the local chord axis based on the identification of the leading and trailing edges. This approach ensured that the extracted sections were normal to the sweep and pre-bend lines.

Blade sections were extracted at intervals of 200 mm, resulting in a total of 223 profiles. However, some profiles were excluded due to evident inaccuracies or because they coincided with the lanyard's position. A subset of 45 sections was deemed sufficient to describe the sweep and pre-bend lines, the local twist angle, and the airfoil sections forming the blade (see Figure 14).

For further details on the resulting distributions, readers can referred to Dubois et al. (2022). The chord and thickness distributions showed very good agreement with the manufacturer's specifications. However, the pre-bend line was found to be less accurate, likely due to blade deformations caused by the lanyard used to stabilize the blade during the scan.





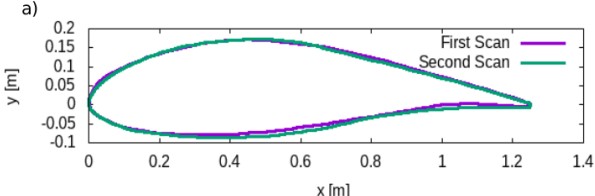

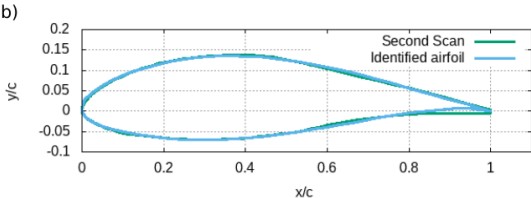

**Figure 15.** Airfoil section accuracy evaluations (more details on the airfoil shapes can be found in Table 3): a) comparison between airfoil shapes from first and second scan. b) Comparison between the identified airfoil shape (i.e. modified Naca 63(3)418) and the second scan shape.

**Table 3.** Airfoil shape parameters.

| Airfoil shape | Max. Thickness [%] | at x/c [%] | Max. Camber [%] | at x/c [%] |
| --- | --- | --- | --- | --- |
| First Scan | 19.84 | 32.90 | 3.83 | 51.60 |
| Second Scan | 20.58 | 35.58 | 3.66 | 47.27 |
| NACA 63(3)418 original | 18.00 | 33.90 | 2.21 | 50.00 |
| NACA 63(3)418 modified | 20.50 | 33.90 | 3.50 | 50.00 |

### 5.3 Accuracy of blade sections and loads

Airfoil sections composing the blade were not provided by the manufacturer, limiting the evaluation of scanning accuracy.
Nevertheless, a partial accuracy assessment was performed using data from the two scanning methods. For this purpose, the blade section at 82% of the rotor diameter was extracted from both scans and compared in terms of shapes.

The blade shapes, shown in Figure 15a, were found to be highly similar, with a maximum difference of 2 mm on the pressure side – well within the measurement accuracy. The reader can find load evaluations at $Re_c = 4.7 \times 10^6$ and $Re_c = 3.6 \times 10^6$, from full chord scale experiments performed at CSTB wind tunnel Neunaber et al. (2022); Braud et al. (2024). Experiments
and simulations were also performed at scale 1:10, $Re_c = 2.10^5$, using the LHEEA's wind tunnel and the ISIS-CFD software Mishra et al. (2024).

The airfoil properties derived from the scans were sufficiently accurate to identify the airfoil family and its associated aerodynamic characteristics. The blade section closely resembles a NACA 63(3)418 profile with modified thickness and camber (see Table 3). For such profiles, a trailing edge stall process is expected as described by Gault (1957). This profile shape also
exhibits local load bi-stability near the maximum lift value as demonstrated full scale Neunaber et al. (2022); Braud et al. (2024). While the presence of stall cells were evidenced numerically at 1:10 scale Mishra (2024).





# 6  Conclusions

A comprehensive meteorological dataset from an operational wind farm, consisting of six 2 MW turbines located in Saint-Hilaire de Chaléons, Pays-de-Loire (commercially operated by VALEMO/VALOREM), has been made available. A meteorological mast was installed at the center of the farm and has collected data over three years, equipped with sonic anemometers at four different heights.

The dataset is further supplemented with radiometer measurements conducted from June 15, 2022, to April 16, 2024, for atmospheric stability analysis. Simultaneously, SCADA data were acquired to provide operational information about the wind turbines, including power production, wind direction, and other key parameters. In addition, the turbine blades were scanned to support aerodynamic simulations.

This unique and comprehensive database has been made accessible to the research community through the AERIS platform.

*Data availability.*

All data were stored on the AERIS platform at this link: https://awit.aeris-data.fr/

The availability of the data sets during the 3-year measurement period are summarized in figure 16.

In addition, all datasets have individual Digital Object Identifiers (DOIs):

- Meteorological mast: https://doi.org/10.25326/589

- Radiometer: https://doi.org/10.25326/600

- SCADA data of turbines T2, T3, T5 and T6: https://doi.org/10.25326/623

- Blade scan: https://doi.org/10.25326/593



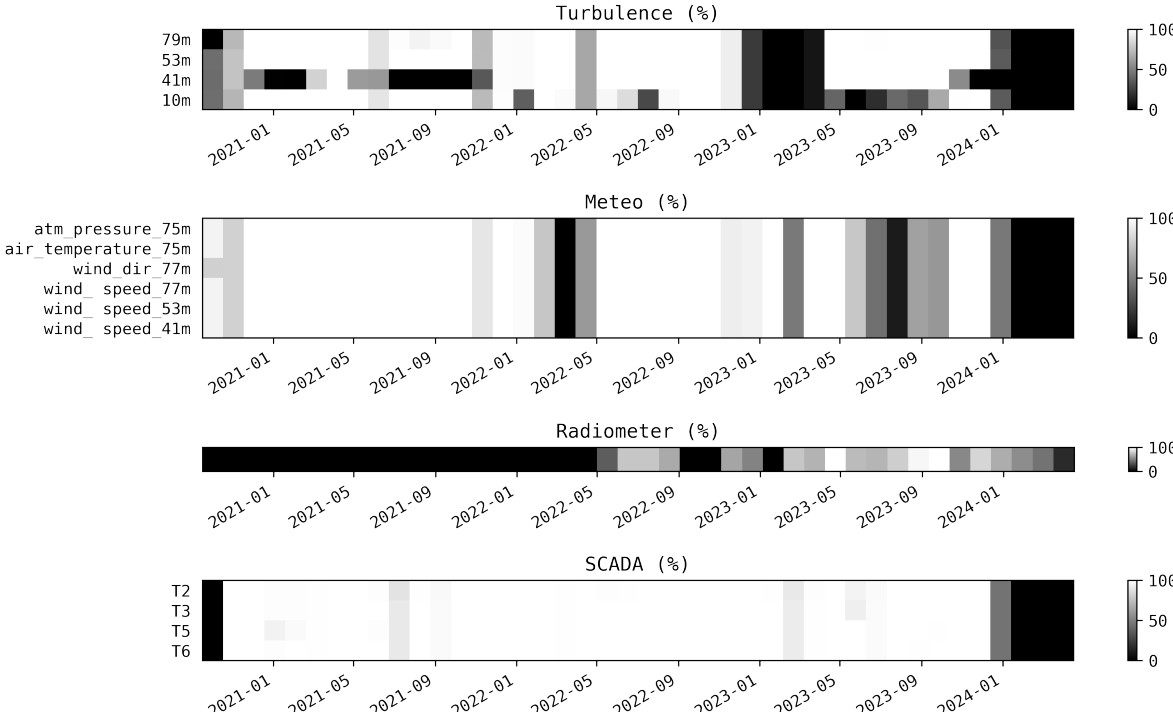

**Figure 16.** Data Availability of data from the sonic anemometers ("Turbulence") and meteorological sensors on the mast ("Meteo"), and from the radiometer ("Radiometer") and the SCADA system of the wind turbines ("SCADA") for the measurement period from December 2020 to January 2024.





*Author contributions.* VALOREM group led by LM has installed the met mast with support from CB and SA for the location and PK for technical issues. PK has installed the sonic anemometers with the VALEMO-Nantes group led by CT. J-FG, EL, PD, PR and J-LA installed, acquired and post-processed the radiometer data within the framework of the ANR MOMENTA project. First treatments of the mast dataset were performed by I. Neunaber under the supervision of PK, SA and CB. Fundings (ANR MOMENTA and ePARADISE projects) were acquired by CB. All authors contributed to writing and editing the manuscript.

*Competing interests.* Sandrine Aubrun is a member of the editorial board of WES.

*Acknowledgements.* The mast and sonic anemometer installation has been carried out within the research project ePARADISE with the funding from ADEME and Pays-de-Loire region (grant no. 1905C0030). IN and MS Belakhadar were financially supported within the research project ANR MOMENTA (grant no. ANR-19-CE05-0034). The authors thank CSTB for providing two sonic anemometers and for their technical help to replace two of them during this campaign. We thanks MS Belakhadar for his contribution to the dataprocessing during
his master internship. The authors would also like to thank B. Conan (Researcher at Centrale Nantes) for his help in the data post-processing regarding comparison with the ERA5 database.





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
