# Peer review of "Three-year database of atmospheric measurements combined with associated operating parameters from a wind farm of 2MW turbines including rotor geometry"

_Wind Energy Science, 2025_

## Referee Comment (RC2)

**Review Comment**

March 10th, 2025

Title: Three-year database of atmospheric measurements combined with associated operating parameters from a wind farm of 2MW turbines and rotor geometry

Author(s): Caroline Braud, Pascal Keravec, Ingrid Neunaber, Sandrine Aubrun, Jean-Luc Attie, Pierre Durand, Philippe Ricaud, Jean-François Georgis, Emmanuel Leclerc, Lise Mourre, and Claire Taymans

MS No.: wes-2025-12

MS type: Research article

Iteration: Initial submission

**General comment**

This manuscript explains open database obtained by 3 years' measurement campaign in an onshore wind farm, which include wind speed/direction and turbulence components measured by cup and sonic anemometer, temperature measured by radiometer, SCADA data and blade shape obtained by 3D scan. This kind of database is very important for future research and will be contribute to industry. However, only introducing or explaining database is not enough for scientific journal paper. Although it's great effort and there are some interesting topics, it should be added more detail explanation about data quality, scientific insight, new findings etc.

In conclusion, a reviewer suggests MAJOR REVISION of this manuscript.

**Specific comment**

| Clause/ Subclause | Line number | Comments |
|---|---|---|
| 1 | Overall | There is no literatures review. Although the authors quote 5 references, all these are not journal papers. Because this is research article, the authors should refer not only industry work but also journal paper. The authors should explain about previous researches, differences between previous researches and this manuscript etc. |
| | 18-36 | Although, the authors raise 4 originalities of present database, it is hard to understand the originality. "Operational Wind Turbine (SCADA) Data" already exists, as |

| | | |
|---|---|---|
| | | the authors explained.

"Measurement of Wind Properties": Sonic anemometer is commonly used for boundary layer and turbulence structure research and open data might exist.

"Expansion of the Database Applications" and "Future Benchmarking" are about future work and not explaining originality of database in this manuscript.

The authors have to explain the originality of present database more clearly. |
| | 30 | It is hard to understand the meaning of "same wind farm environment". Is that same meaning of "same wind farm"? |
| | 31 | "AERISwebsitehttp://..."
Need space between website and the URL. It is suggested to put URL in brackets. |
| 1.1 | 42 | Equivalent to 4D -> 3.8D |
| | Table 1 | Longitude -1.9XXX°W is correct? Or typo of 1.9XXX°W or -1.9XXX°E? |
| 2 | Figure 2 | It is suggested to align vertical axis of figure 2-(a) and 2-(b). |
| | Figure 3 | Image resolution of figure 3-(a) and 3-(b) are bad. It should be replaced by higher ones. Also, figure should be quoted and explained in somewhere in main body. |
| | 61-69 | Flow distortion effect caused by met mast is important to evaluate reliability of wind measurement. At least, mounting direction of each anemometer should be mentioned. Adding figure of x-y planes, explanation of length of mounting booms, frequency of calibrations, standard or guidelines followed are preferable. |

| | 69 | The date when sonic anemometers were replaced should be explained. |
|---|---|---|
| | 72 | In general, 10-min statistics are used especially for turbulence parameters. Why the authors decided to compute 1-hour statistics? Need explanation. |
| | 86-87 | The authors say "Statistical convergence is not acceptable (i.e., [105° −150°]).", however, the reasons are not clearly mentioned. Need explanation. |
| | 88, 89 | Is language "wake interactions" correct? Or is it just mean "wake effect"? If there is interaction, the author need to explain more details about the phenomena. |
| | 89-90 | Although the authors say "The footprint... less sensitive to altitude".
It is hard to understand if this sentence is correct only from figure 5. Add more explanation about the reason that the authors think so in text or figure. |
| 3 | 101-102 | "The vertical resolution of the profiles..."
The explanations of altitudes such as "atmospheric boundary layer, lower free troposphere, etc." are very ambiguous.

It should be more clearly explained the relation of altitude and resolution as numeric. |
| 4 | Figure 8. | Add reference if this figure is already published as somehow (e.g. report, webpage etc.). |
| | Figure 8. | Caption is incorrect. It must be "(10) yaw bearing, (11) nacelle". |
| | Table 2 | "Nacelle height" should be "hub height". |

| | | |
|---|---|---|
| | Table 2 | What does the meaning of "Average weight"? Also, "rotor without blades" is "hub" and "nacelle without rotor" is "nacelle", in general. Is there any reason that the authors call that way? |
| | 114, Title | Title is "Description of the Wind Turbine" and seems explain about only wind turbine. However, main subject of this manuscript is database. It is suggested to change title as "Description of the SCADA data" or "Description of the Wind Turbine and SCADA data" |
| | 120 | Although the authors say that "All wind turbine are equipped SCADA...", the authors also say that "SCADA data from four of the six turbines are included ..." in line 12. It should be clearly explained that which tubines' SCADA data is available in present database. |
| | Table 2 | It is suggested to add cut-in, rated and cut-out wind speeds, as general information. |
| | 130 | What does "azimuth error" mean here? Is that same meaning of "yaw misalignment"? |
| | 129-131 | The expression of this sentence is very ambiguous and not clear if data would be provided by present manuscript is corrected as black-box or couldn't correct due to lack of information.

More clear explanation is needed. |
| | 137,138 | rotation -> rotational speed or rotational frequency |
| 5 | Figure 10, 11 | Figure 11 is referred earlier than figure 10 in main text. The authors should reconsider the order of those figures. |

| | | |
|---|---|---|
| 5.1.1 | 158 | Is "obscured by the mast" intent "obscured by the tower"? If so, it should be corrected. |
| | 167、216 | Is "82% of the rotor diameter" correct? Or does it intent "82% of blade length" |
| | 175 | (see Figures 12a) and b) -> (see Figures 12a) and b)) |
| | Figure 12 | Image resolution of figure 12-(a), 12-(b) and 12-(c) are bad. It should be replaced by higher ones. |
| | 155, 179 | It is suggested to mentions about weather condition such as wind speed for day of scan work. |
| | 192 | mast -> tower |
| 6/Abstract | Overall | It is difficult to understand originality, new findings etc. of this manuscript. The authors should reconsider of these clauses based on reviewer's comments and its modification.

Needless to say, general comment and some of comment on clause 1 are highly important. |

---

## Author Comment (AC1)

**WES-2025-12 - Response to Reviewer 1**
*(The reviewer's comments are in italics)*

The authors thank the Reviewer for their time and feedback on our manuscript. We address their concerns below. The Reviewer's comments are in *italics*. Our replies follow each comment. Changes made to the manuscript for Reviewer 1 are highlighted in purple. Author initiated changes are in turquoise.

**Suggested Revisions:**

1. ***data set Format and Organization****: Add more information about the data formats used. Consider including a summary paragraph on data set organization (e.g., "data set XX is stored in NetCDF format with standardized metadata following XYZ guidelines.").*

    **Reply:** A summary paragraph on data organization and format has been added at the end of the manuscript in Chapter 9 as follows:
    "**Section 8 Data-set format and organization**

    - Meteorological mast data sets are stored in CSV format and organized in three directories: "data availability", "meteo", and "turbulence". The "data availability" directory provides availabilities of the meteorological data sets in CSV format (also shown in Figure 16). The "meteo" directory contains atmospheric data from cup anemometers, a wind vane, a thermometer and an atmospheric pressure sensor. The "turbulence" directory contains the sonic anemometer measurements at the four meteorologic mast heights detailed in the README file.

    - Radiometer data set are stored in NetCDF format and organized in four directories: The CMP.TPC folder contains atmospheric temperature profile data collected over time. The file includes time-indexed measurements across 93 altitude layers, along with metadata and rain flag. The HPC folder contains vertical profiles of absolute and relative humidity collected over time across the atmospheric layers. The data includes metadata on measurement conditions and auxiliary quality indicators such as rain detection. The IWV folder contains time-resolved measurements of Integrated Water Vapor (IWV), along with viewing geometry, retrieval method, and rain flag. The MET folder contains NetCDF files of meteorological measurements (observed at 3 m agl) including pressure, temperature and relative humidity at a temporal resolution of 1 s. Each entry corresponds to a single point in time and includes metadata such as rain status and integration details.

    - SCADA data sets are stored in CSV format and organized in three directories: the "DATA" directory contains the CSV files, the "AVAILABILITY" directory provides availabilities of SCADE data (also shown in Figure 16 of the manuscript) and "MANUFACTURER-INFORMATION" contains the power and thrust curves. For reasons of confidentiality, the data is available for four of the six turbines only. More details on the data organization can be found in the README.txt file.

- The blade scan data can be read using the CSV format as performed in the provided python code "plot-3DBlade.py". The README.txt file explain the content of the directory, including filenames "Rxxm.txt" which contain the 2D airfoil profiles composing the 3D blade."

2. *Please state why some data is unavailable, e.g., SCADA from the two remaining turbines. Was it a requirement from the data provider or something else?*

   **Reply:** This has been specified now in the new Section 8:

   "For reasons of confidentiality, the data is available for four of the six turbines only."

3. ***Terminology Consistency****: Table 2 lists "nacelle height" as 80 m, while the text states "hub height" is 80 m. Please clarify this and ensure consistent terminology.*

   **Reply:** "Nacelle" has been replaced by "hub" in Table 2.

4. ***AERIS Platform****: Provide additional information on the AERIS platform. For example, does it guarantee continuous access for many years into the future?*

   **Reply:** Aeris has existed for 10 years and relies on French ministry engagements (CNRS, CNES, Météo France, . . . ). It is part of four data centers and services that share a certain number of tools which, in case of permanent or temporary failure, can take over data storage and distribution. One of these data centers (ESPRI) is Core Trust Seal certified. The SEDOO data center, where the present data are stored, has introduced a high quality service around data conservation, and is in the process of obtaining of the same certification as ESPRI. SEDOO is a data service dedicated to developing tools for storing, managing, processing, and sharing environmental scientific data. SEDOO supports regional data management needs, contributes nationally through its involvement in the AERIS and ODATIS data hubs and long-term environmental monitoring services, and participates internationally in major scientific programs and multidisciplinary measurement campaigns (`https://www.sedoo.fr`).

5. ***Figure and Table Improvements:***

   (a) *Figure 5 appears grainy; please improve the image quality.*

   (b) *Figures 4, 5, 6, 7, 9, and 16: Add subplot labels (a), (b), (c), etc., unless they are considered single plots.*

   (c) *Table 3: Add more detailed descriptions of columns and rows. Consider explaining terms such as "NACA 63(3)418," "Max. Camber [%]," and "at x/c [%]," and*

*perhaps include an illustration of camber and thickness (add to one of the previous figures if possible).*

**Reply:**

(a) The image resolution of Figure 5 has been improved.

(b) Subplot labels have been added.

(c) The caption of Table 3 has been extended to explain terms in the table:
"Airfoil shape characteristics: The columns contain the airfoil maximum thickness and its location in percentage of the chord, and the maximum camber and its location in percentage of the chord . The two first rows of this table show the airfoil properties for the two scanned cases, while the third row correspond to the original (not modified) NACA profile shape, namely a NACA 63(3)418 profile, as a reference. The fourth row corresponds to a NACA 63(3)418 profile modified in such a way that it fits the scanned data."
The original NACA profile is an airfoil shape referenced here: `https://m-selig.ae.illinois.edu/`. The URL has been added (footnote 8). To avoid overloading figures, a clear definition of an airfoil camber and thickness has been added in footnotes 9 and 10:
9: "The thickness is the maximum difference between the upper and lower airfoil surfaces divided by the chord length"
10: "The mean camber line is an imaginary line which lies halfway between the upper surface and lower surface of the airfoil and intersects the chord line at the leading and trailing edges"

6. ***Acronyms and Notation:*** *Define CSTB, LHEEA, ISIS-CFD, NACA 63(3)418, and URANS, and provide references where possible.*

**Reply:** These acronyms and notations have been added in the manuscript. CSTB is a French research center (Centre Scientifique et Technique du Bâtiment; footnote 5), LHEEA is a French research laboratory (Laboratoire de recherche en Hydrodynamique, Énergétique et Environnement Atmosphérique; footnote 2), ISIS-CFD is a CFD solver developed and maintained by the LHEEA laboratory and sold by NUMECA via the FINE$^{TM}$/Marine Suite (footnote 6), and NACA 63(3)418, as explained above, refers to a profile type from the NACA series. U-RANS stands for Unteady Reynolds-Averaged Navier-Stokes equations (footnote 3).

7. *Clarifications and Formatting:*

   (a) *Line 76: Add a space between the number and unit ("79 m").*

   (b) *Line 236: Consider moving the last line up to align with the preceding paragraph.*

   (c) *Explain if "hNN" refers to height above "normal null," as this may not be self-evident in the wind energy community.*

(d) *Introduce and clarify the notation "$Re_c$."*

**Reply:**

(a) This has been corrected.

(b) This has been corrected.

(c) This is the height above sea level. It has been renamed by $h_{asl}$ for clarity.

(d) $Re_c$ is the airfoil chord based Reynolds number defined as $Re_c = Uc/\nu$ with $U$ the free-stream velocity, $c$ the airfoil chord and $\nu$ the kinematic viscosity. It is well known in the aerodynamics community. It has been clarified in the manuscript in footnote 4.

**Data File Review:**

1. *Meteorological Mast data set(s):*

   (a) **data_availability folder**: *Format: .csv files. Missing README. Files open correctly using Pandas.*

   (b) **meteo folder**: *Format: .cvl file (tab-separated ASCII). It is fair README, but it could describe more about the data formatting and standards (if any). Files open correctly using Pandas.*

   (c) **Turbulence folder**: *Format: .csv files (one per height). Good README. Files open correctly using Pandas.*

   **Reply:** We have now added a README file in the data_availability folder.

2. ***Radiometer data set(s):***

   (a) *General Issues:*

      i. *"readme_v1.rtf" contains a netCDF header dump but lacks a general description of folders, files, and formats.*

      ii. *The "IWV" and "MET" folders are empty and undocumented.*

      iii. *The "Preliminary_Version" folder contains HPC, CMP, and IWV netCDF files, but their purpose is unclear.*

   (b) *File-specific Notes:*

      i. *"lire_cmp_tpc.py": Python script to parse files. Could include more details about the purpose and docstrings for the functions.*

      ii. *"CMP_TPC_report.csv": List of days with missing or present netCDF files (CMP_TPC).*

iii. "CMP.TPC" folder: Daily netCDF files (opens fine using xarray, but missing README apart from readme_v1.rtf).

iv. "HPC" folder: Daily netCDF files (opens fine using xarray, but missing README apart from readme_v1.rtf).

**Reply:**

(a) General Issues:

  i. We have now added a README file to each folder (CMP.TPC, HPC, IWV, and MET, as described above), providing a description of the NetCDF file content.

  ii. The corresponding files are now uploaded with a descriptive README file.

  iii. The "Preliminary Version" folder was removed and replaced by V1 (Version 1) which is the latest version of the data.

(b) File-specific Notes:

  i. The python code "lire cmp tpc.py" has been replaced by "lire cmp tpc v1.py" which includes more details about the purpose of the functions. This is an example code to read the content of the CMP.TPC NetCDF files.

  ii. Yes, but for all files in each folder CMP.TPC, HPC, IWV, and MET.

  iii. We have now added a README file to each folder, providing a description of the NetCDF file content.

  iv. We have now added a README file to each folder, providing a description of the NetCDF file format used.

3. **SCADA data set(s): DATA folder:**

   (a) *Format: Excel file*

   (b) *Good README*

   (c) *Files open correctly using Pandas (and Excel).*

   **Reply:** The format of these data sets has been modified in reply to Reviewer 3. They are now split into two files in CSV format, and we added a python code example to check the readability. The README file has been modified accordingly.

4. **Blade Geometry data set(s):**

   (a) *Text files named RNNm.txt with NN being numbered from 00 to 44. Each file contains three columns. I assume "x," "y," and "z," but there are no headers, so I have to guess.*

(b) *The README contains useful information but lacks basic stuff, like file formats/standards description and information about who the "Authors" (people/institution(s)) of the data set are.*

**Reply:** We added columns titles in the files (xb,yb,zb) in accordance with the data-paper. Authors were added in AERIS. The README file has been improved. A python code to plot the 3D blade has been added.

---

## Author Comment (AC2)

We thank the Reviewer for their time and feedback on our manuscript. We address their concerns below. The Reviewer's comments are in *italics*. Our replies follow each comment. Changes made to the manuscript for Reviewer 2 are highlighted in blue. Author initiated changes are in turquoise.

*This manuscript explains open database obtained by 3 years' measurement campaign in an onshore wind farm, which include wind speed/direction and turbulence components measured by cup and sonic anemometer, temperature measured by radiometer, SCADA data and blade shape obtained by 3D scan. This kind of database is very important for future research and will be contribute to industry. However, only introducing or explaining database is not enough for scientific journal paper. Although it's great effort and there are some interesting topics, it should be added more detail explanation about data quality, scientific insight, new findings etc.*

*In conclusion, a reviewer suggests MAJOR REVISION of this manuscript.*

**Specific Comments**
**Section 1**

- ***Overall*** *- There is no literatures review. Although the authors quote 5 references, all these are not journal papers. Because this is research article, the authors should refer not only industry work but also journal paper. The authors should explain about previous researches, differences between previous researches and this manuscript etc*

  **Reply:** The present manuscript is a data description paper, not a research article. The co-authors believe that they have properly followed the journal recommendations on data description papers. Please refer to the Wind Energy Science webpage describing the manuscript type: "Data description papers describe original and FAIR research data, and the planning, instrumentation, and execution of experiments, collection or generation of data. Articles may describe field or lab-scale observational data, simulation data, or combinations thereof. Although examples of data outcomes may prove necessary to demonstrate data quality, extensive interpretations of data – i.e. detailed analyses as an author might report in a research article – remain outside the scope of this manuscript type. `https://www.wind-energy-science.net/about/manuscript_types.html`"

- ***Ll. 18-36*** *- Although, the authors raise 4 originalities of present database, it is hard to understand the originality.*
  *"Operational Wind Turbine (SCADA) Data" already exists, as the authors explained. "Measurement of Wind Properties": Sonic anemometer is commonly used for boundary layer and turbulence structure research and open data might exist. "Expansion of the Database Applications" and "Future Benchmarking" are about future work and not*

*explaining originality of database in this manuscript.*
*The authors have to explain the originality of present database more clearly.*

**Reply:** As mentioned in the introduction, access to turbine data (SCADA information, blade geometry, ...) together with meteorological and turbulence measurements is only possible through collaborations between academia and industry. These data are generally restricted due to confidentiality reasons. Compared to other available data sets, the volume of the meteorological mast and SCADA data sets is more extensive, the blade geometry is provided, and temperature and humidity profiles derived from radiometer measurements are included.

- **L. 30** - *It is hard to understand the meaning of "same wind farm environment". Is that same meaning of "same wind farm"?*

  **Reply:** Yes, this is the same wind farm site. It has been clarified (ll. 29): "This initial database serves as the foundation for further data sets generated within the same wind farm site."

- **L. 31** - *"AERISwebsitehttp://..." Need space between website and the URL. It is suggested to put URL in brackets*

  **Reply:** This has been corrected.

**Section 1.1**

- **L. 42** - *Equivalent to 4D → 3.8D*

  **Reply:** This has been corrected.

- **Table 1** - *Longitude -1.9XXX°W is correct? Or typo of 1.9XXX°W or - 1.9XXX°E?*

  **Reply:** This has been corrected, thank you for pointing it out.

**Section 2**

- **Figure 2** - *It is suggested to align vertical axis of figure 2-(a) and 2-(b).*

  **Reply:** This has been corrected.

- **Figure 3** - *Image resolution of figure 3-(a) and 3-(b) are bad. It should be replaced by higher ones. Also, figure should be quoted and explained in somewhere in main body.*

  **Reply:** Figure 3 is now referenced in the text, and the image resolution has been improved.

- **Ll. 61-69** - *Flow distortion effect caused by met mast is important to evaluate reliability of wind measurement. At least, mounting direction of each anemometer should be mentioned. Adding figure of x-y planes, explanation of length of mounting booms, frequency of calibrations, standard or guidelines followed are preferable.*

  **Reply:** This has been updated with the following text:

"The lower levels' sonic anemometers (10 m, 41.8 m, , and 53.5 m) are mounted on a 2 m boom pointing to 302°. This orientation is chosen in order to minimize the time spent in the wake of the mast, as 120° wind does not occur often as shown by the wind roses."

- **L. 69** - *The date when sonic anemometers were replaced should be explained.*

  **Reply:** This has been updated with the following text:

  "During the field experiment, the top (December 2020) and 41.8 m (February 2022) anemometers were replaced with two Gill WindMaster Pro sonic anemometers due to failure of the installed one."

  r

- **L. 72** - *In general, 10-min statistics are used especially for turbulence parameters. Why the authors decided to compute 1-hour statistics? Need explanation*

  **Reply:** 1-hour turbulence statistics were calculated using 20 Hz data sets from the sonic anemometers. This was chosen as a first approach for the data release to resolve large scales that dominate the atmospheric energy spectra. For instance, this is used to compute atmospheric stability. It can be adapted and reprocessed later, depending on the user demand.

- **Ll. 86-87** - *The authors say "Statistical convergence is not acceptable (i.e., [105°-150°]).", however, the reasons are not clearly mentioned. Need explanation*

  **Reply:** For these wind orientations, the sonic anemometer is in the wake of the meteorologic mast, preventing interpretable results. This has been clarified in the manuscript (ll. 86):

  "Green dots represent wind directions where the met mast is not affected by turbine wakes and where statistical convergence is acceptable (i.e., $[90° - 105°]$ ; $[150° - 240°]$ and $[285° - 300°]$). In contrast, black dots indicate wind directions where the met mast is impacted by wind turbine wakes (i.e., $[0° - 90°]$ ; $[240° - 285°]$ and $[300° - 360°]$), or where the sonic anemometer is in the wake of the meteorologic mast (i.e., $[105° - 150°]$)."

- **Ll. 88, 89** - *s language "wake interactions" correct? Or is it just mean "wake effect"? If there is interaction, the author need to explain more details about the phenomena.*

  **Reply:** Yes, it is the same as wake effect. We replaced "wake interaction" by "wake effect" and we clarified that it corresponds to the interaction between a wind turbine wake and the met mast (ll. 86):

  "Green dots represent wind directions where the met mast is not affected by turbine wakes and where statistical convergence is acceptable (i.e., $[90° - 105°]$ ; $[150° - 240°]$ and $[285° - 300°]$). In contrast, black dots indicate wind directions where the met mast

is impacted by wind turbine wakes (i.e., $[0° - 90°]$ ; $[240° - 285°]$ and $[300° - 360°]$) , or where the sonic anemometer is in the wake of the meteorologic mast (i.e., $[105° - 150°]$)."

- **Ll. 89-90** - *Although the authors say "The footprint... less sensitive to altitude". It is hard to understand if this sentence is correct only from figure 5. Add more explanation about the reason that the authors think so in text or figure.*

  **Reply:** Yes, this statement is based on Fig. 5. In this figure, one notices that the turbulence intensity measured for wind directions where the met mast is affected by turbine wakes is not sensitive to the altitude. It is not the purpose of a data paper to interpret the results in a more extensive way.

**Section 3**

- **Ll. 101-102** - *"The vertical resolution of the profiles..." The explanations of altitudes such as "atmospheric boundary layer, lower free troposphere, etc." are very ambiguous. It should be more clearly explained the relation of altitude and resolution as numeric.*

  **Reply:** We replaced the unclear paragraph by the following more detailed paragraph (ll. 104):

  "The first levels of the vertical profiles are 10, 25, 50 and 75 m above ground level (agl). Above this, the vertical resolution of the profiles ranges from 30 to 40 m between 100 m and 1200 m agl, corresponding approximately to the atmospheric boundary layer. Then, up to 10 km agl, the resolution varies between 60 and 300 m."

**Section 4**

- **Figure 8** - *Add reference if this figure is already published as somehow (e.g. report, webpage etc.).*

  **Reply:** The reference, Lebranchu (2016), has been added.

- **Figure 8** - *Caption is incorrect. It must be "(10) yaw bearing, (11) nacelle".*

  **Reply:** This has been corrected.

- **Table 2** - *"Nacelle height" should be "hub height".*

  **Reply:** This has been corrected.

- **Table 2** - *What does the meaning of "Average weight"? Also, "rotor without blades" is "hub" and "nacelle without rotor" is "nacelle", in general. Is there any reason that the authors call that way?*

  **Reply:** These are terms employed by the manufacturer in their documents. We did not measure anything, the information was extracted from documents provided to VALEMO. We kept these terms for consistency with these documents. We however clarify now in the caption of Table 2 that these values are extracted from technical documents provided by the wind turbine manufacturer at its acquisition:

"Characteristics of the wind turbines extracted from technical documents provided by the wind turbine manufacturer at its acquisition."

- **L. 114, Title** - *Title is "Description of the Wind Turbine" and seems explain about only wind turbine. However, main subject of this manuscript is database. It is suggested to change title as "Description of the SCADA data" or "Description of the Wind Turbine and SCADA data"*

  **Reply:** This has been corrected.

- **L. 120** - *Although the authors say that "All wind turbine are equipped SCADA...", the authors also say that "SCADA data from four of the six turbines are included ..." in line 12. It should be clearly explained that which tubines' SCADA data is available in present database.*

  **Reply:** This has been specified in the added Section 8 (ll. 271; color of Reviewer 1): "For reasons of confidentiality, the data is available for four of the six turbines."

- **Table 2** - *It is suggested to add cut-in, rated and cut-out wind speeds, as general information.*

  **Reply:** The information has been added to Table 2.

- **L. 130** - *What does "azimuth error" mean here? Is that same meaning of "yaw misalignment"?*

  **Reply:** Yes, it refers to yaw misalignment. It has been replaced for clarity.

  "Both wind speed and wind direction are corrected for deviations caused by rotor rotation (which also varies according to wind speed and yaw misalignment), although the transfer functions used for these corrections are not provided by the manufacturer and remain unknown."

- **Ll. 129-131** - *The expression of this sentence is very ambiguous and not clear if data would be provided by present manuscript is corrected as black-box or couldn't correct due to lack of information. More clear explanation is needed*

  **Reply:** Anemometers are placed on the nacelle behind the rotor. This induces deviations in the wind speed and yaw measurements that can be removed using a transfer function. This transfer function has been applied in the available data, we however cannot go back to the raw data as the transfer function is unknown because not provided by the manufacturer.

  The text has been updated:

  "Both wind speed and wind direction are corrected for deviations caused by rotor rotation (which also varies according to wind speed and yaw misalignment), although the transfer functions used for these corrections are not provided by the manufacturer and remain unknown."

- **Ll. 137, 138** - *rotation → rotational speed or rotational frequency*

  **Reply:** The related sentences have been reformulated for clarity.

**Section 5**

- **Figure 10, 11** - *Figure 11 is referred earlier than figure 10 in main text. The authors should reconsider the order of those figures.*

  **Reply:** This has been corrected.

**Section 5.1.1**

- **L. 158** - *Is "obscured by the mast" intent "obscured by the tower"? If so, it should be corrected*

  **Reply:** Yes, this has been corrected.

- **Ll. 167, 216** - *Is "82% of the rotor diameter" correct? Or does it intent "82% of blade length"*

  **Reply:** "82% of the blade length", this has been corrected.

- **L. 175** - *(see Figures 12a) and b) → (see Figures 12a) and b))*

  **Reply:** This has been corrected.

- **Figure 12** - *Image resolution of figure 12-(a), 12-(b) and 12-(c) are bad. It should be replaced by higher ones*

  **Reply:** Figure 12-(a) and (b) were originally smaller which explains this bad image quality. To improve the image quality, the only option is to keep the original image size. Figure 12 (c) was of high quality already and was unchanged.

- **Ll. 155, 179** - *It is suggested to mentions about weather condition such as wind speed for day of scan work.*

  **Reply:** The wind speed was 3.57 m/s for the second scan. It has been added.

- **L. 192** - *mast → tower*

  **Reply:** This has been corrected.

**Section 6/Abstract**

- **Overall** - *It is difficult to understand originality, new findings etc. of this manuscript. The authors should reconsider of these clauses based on reviewer's comments and its modification. Needless to say, general comment and some of comment on clause 1 are highly important.*

  **Reply:** Again, this is a data paper, not a research paper (see first response).

**References**

Lebranchu, A.: Analyse de données de surveillance et synthèse d'indicateurs de défauts et de dégradation pour l'aide à la maintenance prédictive de parcs de turbines éoliennes. Traitement du signal et de l'image, Ph.D. thesis, Université Grenoble Alpes, URL `https://theses.hal.science/tel-01503571v2`, 2016.

---

## Author Comment (AC3)

We thank the Reviewer for their time and feedback on our manuscript. We address their concerns below. The Reviewer's comments are in *italics*. Our replies follow each comment. Changes made to the manuscript for Reviewer 3 are highlighted in green. Author initiated changes are in turquoise.

**Data Availability**

1. *I was not able to access the met mast data, unfortunately—there does not seem to be data under the Download tab? Perhaps I am not accessing it correctly.*

   **Reply:** We have checked the data access, which is available today. We also asked AERIS for possible interruptions. They reported us interruptions from April 17th to 21st 2025. No other interruptions are known outside these dates. If the reviewer can provide the access trial day, this would help us to look further. What could happen: the user cannot download if the box "I agree to the data policy" is not checked.

2. *The SCADA data is stored as an .xlsx file. While this works, I would highly recommend providing the data as a CSV file instead. The comma-separated values format is human readable, can be opened in a wider variety of applications, and is generally smaller in terms of storage size. Moreover, xlsx binary files may contain macros that can be exploited to provide unauthorized access, which means that opening downloaded xlsx files can pose a security risk. CSV files don't have this issue, which should make the data more readily available.*

   **Reply:** The format of these data sets has been modified. They are now split into two files in CSV format, and we added a python code example to check the readability. The README file has been modified accordingly.

3. *As noted in the paper, only 4 of the 6 turbines' SCADA data is provided. However, the reason for this is not given. Why are the other two turbines' records not provided? This should be stated clearly in the paper to avoid confusion.*

   **Reply:** This has been specified in the added Section 8 (Color of Reviewer 1): "For reasons of confidentiality, the data is available for four of the six turbines only."

4. *Do the authors have access to the power and thrust curves (as a function of wind speed) of the Senvion MM92 turbines that they can share (alternatively, power coefficient and thrust coefficient as a function of wind speed)? Several lower-fidelity wake models*

*require these power/thrust curves to model the wind turbines, so it would make the data set more useful if they can be provided.*

**Reply:**   Yes, they were already provided in the initial data set. To put forward this information, data sets were reorganized in different folders with a folder dedicated to the power and thrust coefficients: MANUFACTURER-INFORMATION. The README file has been rewritten accordingly.

**Other Comments**

1. *Can the authors provide specifications for the sensing hardware on the met tower? This would be helpful for users to understand the operational ranges, signal to noise ratios, etc of the anemometers and vanes. The authors provide the manufacturer of each in parenthesis, but these are not linked references, so it's not totally clear what the equipment is. Another alternative would be to provide a footnote with a link to the spec sheet for each sensor, similar to the way that the EddyPro software is linked.*

   **Reply:** We added references when available using footnotes 2, 3 and 4 of page 5.

2. *I understand that Fig. 4 shows the wind roses at various heights on the met mast, covering a period from Dec. 2021 to Jan. 2024. I recommend limiting the period to Dec. 2021–Dec. 2023 or Jan. 2022–Jan. 2024 (exclusive), so that an integer number of years is included and winter months are not double-counted in the presented wind roses.*

   **Reply:**   This has been updated, the period is now January 1st, 2021 - December 31st, 2023.

3. *Can the resolution in Figs. 4 and 5 be improved?*

   **Reply:**   The resolution of Figures 4 and 5 has been improved.

---

## Author Response (AR2)

**WES-2025-12 - Response to Editor**

The authors thank the Editor and associate Editor for their time and feedback on our manuscript. We provided the versions of the Thies First Class Advanced anemometers using the following added line (in red in the manupscript):

"Anemometer Version 4.3351.00.000 is used at 41.3 m and version 4.3351.10.000 is used at 53 m and 77 m."